

# Modelling daily water temperature from air temperature for the Missouri River

Senlin Zhu[1], Emmanuel Karlo Nyarko[2] and Marijana Hadzima-Nyarko[3]

[1] State Key Laboratory of Hydrology-Water resources and Hydraulic Engineering, Nanjing Hydraulic Research Institute, Nnajing, China
[2] Faculty of Electrical Engineering, Computer Science and Information Technology Osijek, University J.J. Strossmayer in Osijek, Osijek, Croatia
[3] Faculty of Civil Engineering Osijek, J.J. Strossmayer University of Osijek, Osijek, Croatia

## ABSTRACT

The bio-chemical and physical characteristics of a river are directly affected by water temperature, which thereby affects the overall health of aquatic ecosystems. It is a complex problem to accurately estimate water temperature. Modelling of river water temperature is usually based on a suitable mathematical model and field measurements of various atmospheric factors. In this article, the air–water temperature relationship of the Missouri River is investigated by developing three different machine learning models (Artificial Neural Network (ANN), Gaussian Process Regression (GPR), and Bootstrap Aggregated Decision Trees (BA-DT)). Standard models (linear regression, non-linear regression, and stochastic models) are also developed and compared to machine learning models. Analyzing the three standard models, the stochastic model clearly outperforms the standard linear model and nonlinear model. All the three machine learning models have comparable results and outperform the stochastic model, with GPR having slightly better results for stations No. 2 and 3, while BA-DT has slightly better results for station No. 1. The machine learning models are very effective tools which can be used for the prediction of daily river temperature.

## INTRODUCTION

River temperature is an important factor that can be used to determine the health of an aquatic ecosystem. All aquatic species have a specific river water temperature range that they can tolerate and significant changes in river water temperature might detrimentally affect aquatic species (*Caissie, Satish & El-Jabi, 2007*). For example, studies have shown that high river temperature reduces survival of adult migrating sockeye salmon (*Hinch et al., 2012*). Migration of fish species in a river is probable especially if the maximum weekly water temperature exceeds its temperature tolerance (*Eaton et al., 1995*). A fish species may also perish because of osmoregulatory dysfunction if the weekly water temperature drops below a threshold temperature (*Mohseni, Stefan & Erickson, 1998*). Warmer temperatures are expected to raise mountain stream temperatures, affecting water quality (dissolved oxygen concentrations) and ecosystem health (*Ficklin, Stewart & Maurer, 2013*).

Corresponding authors
Senlin Zhu, senlinzhu@whu.edu.cn
Marijana Hadzima-Nyarko,
mhadzima@gfos.hr

Additionally, water temperature affects many bio-chemical processes present in rivers, such as temperature-dependent metabolism (*Sandersfeld, Mark & Knust, 2017*).

The prediction of river water temperature presents an interesting topic since the water temperature has a significant ecological and economical role (*Grbić, Kurtagić & Slišković, 2013*). Many factors affect river temperature, such as meteorological conditions, river bed conditions, river topography and flow discharge (*Caissie, 2006*). In addition, anthropogenic activities (*Hester & Doyle, 2011*) and the hydrological regime (*Kelleher et al., 2012*; *Lisi et al., 2015*; *Piccolroaz et al., 2016*) also impact river temperature prediction. Thus, it is an invaluable task to have a detailed understanding of the ongoing processes related to the river temperature as well as analyze the different impacting factors (*Hadzima-Nyarko, Rabi & Šperac, 2014*). Meteorological conditions, especially air temperature, wind speed, solar radiation and humidity, are the factors that have the greatest impact because they determine the heat exchange and fluxes taking place at the surface of the river. In regression analysis of river temperature, air temperature is often used as the only independent variable since it can be used as a substitute for the net exchange in heat fluxes that affect the water surface, and also because air temperature estimates the equilibrium temperature of a water course (*Stefan & Preud'homme, 1993*; *Mohseni & Stefan, 1999*; *Webb, Clack & Walling, 2003*; *Caissie, 2006*). Additionally, air temperature is widely measured and more available than full energy budget components. Therefore, it is of great importance to study and explain air-water temperature relationship.

Many water temperature models which have been successfully developed and applied in the past years can be generally categorized into deterministic models and statistical models (*Caissie, 2006*; *Benyahya et al., 2007*; *Cole et al., 2014*). Deterministic water temperature models simulate the spatial and temporal variations of river water temperature based on energy balances of heat fluxes and mass balances of flow fluxes in a water body (*Hebert et al., 2011*). These deterministic models need a great number of input variables, such as river geometry, hydrological and meteorological conditions, and thus are often times impractical and time consuming due to their complexity. Statistical models have been widely used in water temperature predictions because these models are relatively simple and need minimal data inputs. Linear regression models (*Morrill, Bales & Conklin, 2005*; *Krider et al., 2013*), non-linear regression models (*Mohseni, Stefan & Erickson, 1998*; *Van Vliet et al., 2012*), and stochastic models (*Ahmadi-Nedushan et al., 2007*; *Rabi, Hadzima-Nyarko & Sperac, 2015*) have been developed successfully for data relating to different time scales in the past years. Although these statistical models, which link water to air temperatures provide quite simple approaches for water temperature prediction, other statistical models, such as Box Jenkins and non-parametric models (*Benyahya et al., 2007*), and hybrid statistical-physical based models as air2water (*Toffolon & Piccolroaz, 2015*; *Piccolroaz et al., 2016*) can adequately model the water temperature.

Artificial Neural Network (ANN) models have been applied frequently in prediction and forecasting river water temperatures in recent years (*Karaçor, Sivri & Uçan, 2007*; *Sahoo, Schladow & Reuter, 2009*; *Hadzima-Nyarko, Rabi & Šperac, 2014*; *DeWeber & Wagner, 2014*; *Piotrowski et al., 2015*). *DeWeber & Wagner (2014)* developed an ANN ensemble model to predict the mean daily water temperature using air temperature, landform

attributes and forested land cover as predictors. Results showed that under current conditions and future projections of climate and land use changes, the ANN model may prove to be a powerful tool to predict water temperatures, thus it provides valuable information to the management of river ecosystems. *Hadzima-Nyarko, Rabi & Šperac (2014)* generated ANN models and used them to investigate the air and water temperature relationship for the River Drava in Croatia, and concluded that ANN models are suitable for modelling daily mean river temperature.

Though many models have been successfully developed and applied for river water temperature prediction, several key temperature modeling issues have to be noticed, since they may lay the basis for the development of effective river temperature models with medium complexity. (1) The relationship between air temperature and river temperature is non-linear for high or low air temperatures, which has been proved by *Mohseni, Stefan & Erickson (1998)*. When air temperatures are close to or below 0 °C, air and water temperatures are no longer synchronized, leading to a poor relationship between air and river temperatures. The hydrological regime plays a significant role in these cases. (2) River temperature does not respond instantaneously to the variation of air temperature due to thermal inertia, depending on the hydrological regime of the river (*Stefan & Preud'homme, 1993*; *Isaak et al., 2012*; *Toffolon & Piccolroaz, 2015*), thus time lags may need to be considered in air temperature effects for effective water temperature prediction (*Benyahya et al., 2008*). (3) Estimation problems can be induced by spatial and temporal autocorrelation (*Caissie, 2006*; *Benyahya et al., 2007*). (4) Water temperature may be separated into two different components: the long-term periodic component and the fluctuating short-term component. The long-term periodic component can be modeled with simple functions, such as a single invariant sine function, or more complex models, such as Fourier series (*Caissie, El-Jabi & St-Hilaire, 1998*). (5) Many river sites have incomplete data sets even though the amount of observed water temperature data over the world is increasing dramatically (*Webb et al., 2008*). Very few study areas have a complete matrix of sampling sites and years: data may be missing for an entire year at a site or may be incomplete within a year. Incomplete observed data will affect river water temperature prediction depending on the extent and timing of the missing dataset (*Letcher et al., 2016*).

Selection of appropriate model inputs and the suitable time lags have not been intensively investigated in the literature, especially in the case of estimation of river water temperature and other water quality parameters (*Maier & Dandy, 2000*). The selection process is usually based on a priori knowledge about water science and the dependence of water temperature on different meteorological factors. *Hadzima-Nyarko, Rabi & Šperac (2014)* tested six different multilayer perception ANN models using the day of year and time lags of the daily mean air temperature as inputs. Suitable model structures were obtained by performing cross-validation. A suitable radial basis function model was also obtained in a similar manner.

*Piotrowski et al. (2015)* compared various ANN models for river water temperature predictions (multi-layer perceptron, product-units, adaptive-network-based fuzzy inference systems and wavelet neural networks) and nearest neighbor method for short term river water temperature predictions. The results showed that simple and popular multilayer
perceptron neural networks are not outperformed by more complex and advanced models in most cases. *Grbić, Kurtagić & Slišković (2013)* applied a Gaussian Process Regression (GPR) model for daily mean water temperature prediction for the river Drava in Croatia. The proposed approach was compared to traditional modelling approaches, including linear regression models, logistic models and stochastic regression models. Decision trees represent a supervised learning method approach popular in machine learning which can be used for predictive modeling. However, to the best knowledge of the authors, it has never been applied in water temperature modelling. This short overview, in which scientists have attempted to predict river water temperature, found a lack of comparison of methods based on ANN, GPR, decision trees and traditional modelling approaches. In some cases, ANN models provided better results, while in other cases, some other regression models did. With this in mind, various machine learning models were developed to explain the relationship between the daily air and river water temperature for the Missouri River by addressing most of the issues listed above.

## MATERIALS & METHODS

### Study area

The Missouri River flows more than 3,680 kilometers from Three Forks at Montana to St. Louis at Missouri. It is one of the largest tributaries of the Mississippi River. It is an economic lifeline for the watershed, and supports industry, agriculture and outdoor recreations for a long time range. It also provides habitat for wildlife and drinking water for the residents in the whole watershed. However, due to excess uses, the Missouri River encounters some ecological issues. The Missouri River Recovery Program, aiming to replace lost habitat for threatened and endangered wildlife, such as the least tern, pallid sturgeon, and piping plover has been undertaken by the US Army Corps of Engineers (USACE). Water temperature modelling is a key habitat factor in this program because all aquatic organisms need an appropriate temperature to survive. Previously, deterministic water temperature models (HEC-RAS models) have been developed for the Missouri River Recovery Management Plan and Environmental Impact Statement Analysis (*Zhang & Johnson, 2016*). Due to limited observed data, water temperatures for all inflow boundaries were generated using multiple linear regression approach, which was used to predict river water temperature from air temperatures (*Zhang & Johnson, 2016*). In this study, machine learning methods were used to further investigate air-water temperature relationships in the Missouri River. Three river stations from the upstream to downstream Missouri River were selected. The geographic locations of the river stations and the corresponding meteorological stations are presented in Fig. 1. The detailed information about these three stations is summarized in Table 1. Figure 2 shows the time series of the daily mean air temperatures, corresponding water temperatures and available flow discharges for the three stations. Air temperatures are derived from the US Environmental Protection Agency (USEPA) website. Flow discharges are derived from the US Geological Survey (USGS) website. Water temperatures are provided by USACE Omaha and Kansas City Districts.

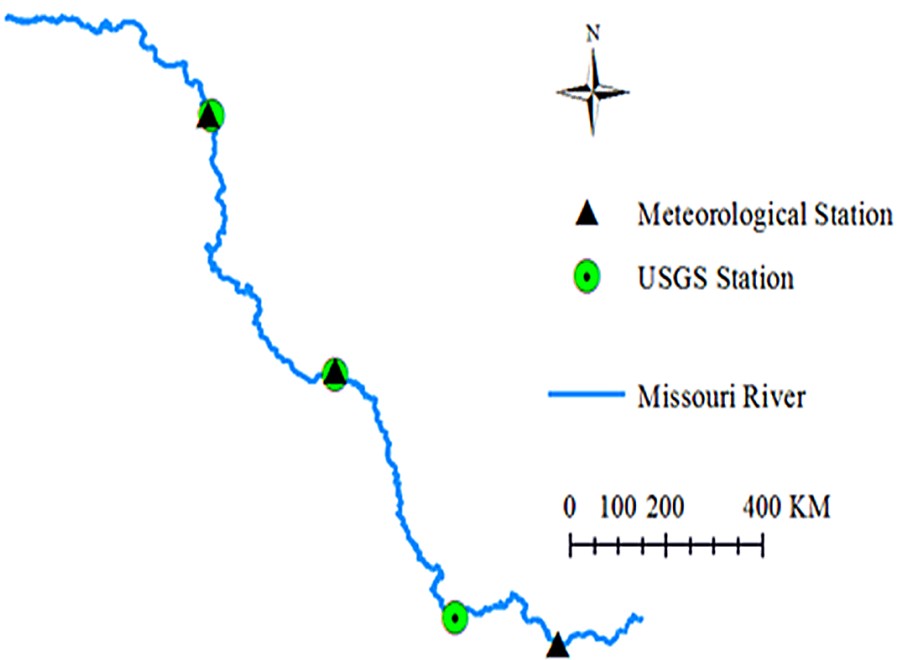

**Figure 1  Geographic locations of the three river stations and meteorological stations.**

**Table 1  Detailed information about the three stations used in this study.**

| Station No. | Station name | Long. | Lat. | Water temperature period | Meteorological station |
|---|---|---|---|---|---|
| 1 | Apple Creek Nr Menoken, ND USGS 06349500 | −100°39′25″ | 46°47′40″ | 2010/1/1–2013/12/31 | ND320819 |
| 2 | Missouri River at Yankton, SD USGS 06467500 | −97°23′38″ | 42°51′58″ | 2010/10/1–2013/7/16 | SD726525 |
| 3 | Missouri River at Kansas City, MO USGS 06893000 | −94°35′17″ | 39°06′43″ | 2007/5/9–2013/12/30 | MO724458 |

### Standard water-air temperature regression models

Three standard models, which describe the relationship between air and water temperatures, are compared to the proposed machine learning modelling approaches. These models are the linear regression model, a non-linear regression model and a stochastic regression model.

*Linear regression model*

For linear regression models, where one dependent variable exists, water temperature is modelled with linear functions. Linear regression models provide a first order estimation of the sensitivity of water temperature on air temperature (*Webb & Nobilis, 1997*; *Kothandaraman, 1971*). This simple model is shown in Eq. (1):

$$T_w(t) = A + BT_a(t) + \varepsilon(t) \tag{1}$$

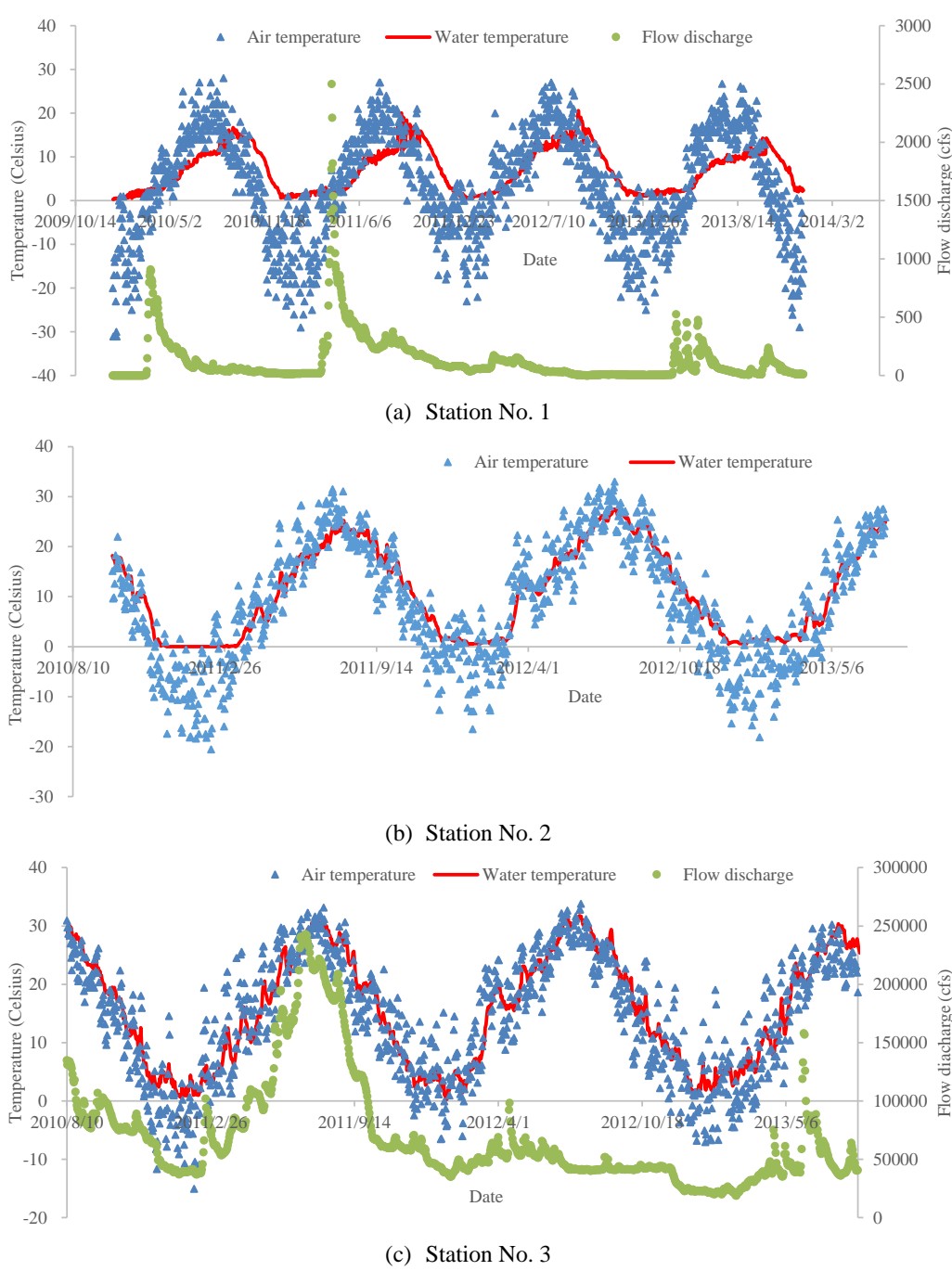

(a) Station No. 1

(b) Station No. 2

(c) Station No. 3

**Figure 2** **Time series of air temperatures, water temperatures and flow discharges for the studied three stations: (A) time series of daily averaged air temperatures, corresponding water temperatures and flow discharges for Station No. 1; (B) time series of daily averaged air temperatures, and corresponding water temperatures for Station No. 2 (no available flow data for station No. 2); (C) time series of daily averaged air temperatures, corresponding water temperatures and flow discharges for Station No. 3.**

where $T_w(t)$ is water temperature for a given day, $T_a(t)$ is air temperature for the same day as water temperature, $A$ and $B$ are regression parameters, and $\varepsilon(t)$ is an error term.

### Non-linear regression model

Whether water and air temperature relationship is linear is a key point for linear regression models, and this assumption has been questioned. For example, *Mohseni, Stefan & Erickson (1998)* found a non-linear relationship between water and air temperature, and they developed a non-linear regression model for water temperature prediction based on a logistic S-shaped function. The non-linear regression model proposed by *Mohseni, Stefan & Erickson (1998)* is shown in Eq. (2):

$$T_w(t) = \mu + \frac{\alpha - \mu}{1 + e^{\gamma(\beta - T_a(t))}} \tag{2}$$

where $\alpha$ is a coefficient which estimates the highest water temperature, $\mu$ is a coefficient which estimates the minimum water temperature, $\beta$ represents the air temperature at the inflexion point, $\gamma$ represents the steepest slope of the logistic S-shaped function.

### Stochastic model

Stochastic models are normally based on a stochastic time-series analysis function which can predict water temperatures using air temperatures (*Caissie, El-Jabi & St-Hilaire, 1998*). In stochastic models, water temperature is generally modeled as a function of time consisting of two entirely different components, a short-term residual component and a long-term component which is periodic in time. There are several forms of stochastic models. One form is used by *Hadzima-Nyarko, Rabi & Šperac (2014)*:

$$T_w(t) = a + b\sin\left[\frac{2\pi}{365}(t + t_0)\right] + \beta_1 T_a(t) + \beta_2 T_a(t-1) + \beta_3 T_a(t-2) \tag{3}$$

where $a, b, t_0, \beta_1, \beta_2, \beta_3$ are regression coefficients.

## Water-air temperature relationship models using machine learning procedures

Three different machine learning procedures were implemented for modeling the water-air temperature relationship and compared, namely: aNN, GPR and Bootstrap Aggregated Decision Trees (BA-DT). The models were required to determine the functional dependence of the water temperature $T_w(t)$ on the input variables $t$, $T_a(t)$, $T_a(t-1)$ and $T_a(t-2)$. The aforementioned input variables or predictors were selected based on the results obtained in *Hadzima-Nyarko, Rabi & Šperac (2014)*.

### Artificial neural networks

Artificial Neural Networks are often used to model complex nonlinear functions from observations, which is their biggest advantage. The most popular ANN is the multi-layer backpropagation network, with a basic structure generally consisting of three distinctive layers: the input layer, one or more hidden layers, and the output layer. In such a network, data travels forward through the layers. Data is introduced to the model through the input layer, then it is processed in the hidden layers, and finally output through the output layer. Every layer is made up of nodes or neurons which are linked to other neurons in the

proceeding layer. With the exception of the neurons in the input layer, all the neurons in the other layers are made up of several components: an activation function, an offset or bias and weights. Common activation functions are *logsig* and *tansig* functions, which are sigmoid non-linear functions, and *purelin* is a linear function.

The training of the network is performed by comparing the desired or measured outputs with the network or model outputs. The difference between these values is then used to adjust the neuron weights. This process is repeated for a sufficiently large amount of training cycles until the desired network output accuracy is achieved.

The stability of ANN model strongly depends on the number of neurons in the hidden layers. A review on approaches to determine the suitable number of hidden neurons within a neural network was performed by *Gnana Sheela & Deepa (2013)*, where they pointed out that the random selection of the number of hidden neurons may cause errors for the constructed models by either overfitting or underfitting. In this paper, the optimal number of neurons for the ANN model of each station was determined by performing 10-fold cross validation using the training (or calibration) dataset of each station. For a given station, the optimal number of neurons was determined by gradually increasing the number of neurons from 1 to 10 in the single hidden layer of the ANN model and calculating the mean cross validation error for each given ANN structure. The results of this procedure are displayed in Fig. 3. For the studied three stations, the optimal number of hidden neurons are respectively 3, 5 and 6.

### Gaussian process regression

Gaussian Process Regression models are not new in hydrology (*Grbić, Kurtagić & Slišković, 2013*). GPR is a full Bayesian learning algorithm which has been used in diverse applications such as experiment design, multivariate regression, model approximation (*Rasmussen & Williams, 2006*; *Girard et al., 2003*; *Quiñonero-Candela & Rasmussen, 2005*). A process is referred to as a Gaussian processes if it is assumed that the joint probability distribution of model outputs is Gaussian. Compared to other machine learning methods, the advantage of GPR is that it combines various machine learning tasks, which include model training, uncertainty estimation and hyperparameter estimation. In GPR, it is assumed that the output variable measurements of $y$ can be represented as

$$y = f(\boldsymbol{x}(k)) + \varepsilon \tag{4}$$

where $\boldsymbol{x}$ represents measurements of input variables, $\varepsilon$ represents noise with a Gaussian distribution and variance $\sigma_n^2$ and $f$ represents the unknown nonlinear function to be modelled.

The space of functions has a prior probability which is defined as a Gaussian process having mean $m(\boldsymbol{x})$ and covariance function $\text{cov}(\boldsymbol{x}, \boldsymbol{x}')$ (*Rasmussen & Williams, 2006*). For a given sample of input variables $\boldsymbol{x}_*$, the output variable is predicted using the predictive probability distribution $p(y_* | \boldsymbol{X}, \boldsymbol{y}, \boldsymbol{x}_*)$ with mean and variance:

$$\begin{aligned} \hat{y}_* &= m(\boldsymbol{x}_*) + \mathbf{k}_*^{\mathrm{T}} \left(\mathbf{K} + \sigma_n^2 \mathbf{I}\right)^{-1} \left(\boldsymbol{y} - m(\boldsymbol{x}_*)\right) \\ \sigma_{y_*}^2 &= k_* + \sigma_n^2 - \mathbf{k}_*^{\mathrm{T}} \left(\mathbf{K} + \sigma_n^2 \mathbf{I}\right)^{-1} \mathbf{k}_* \end{aligned} \tag{5}$$

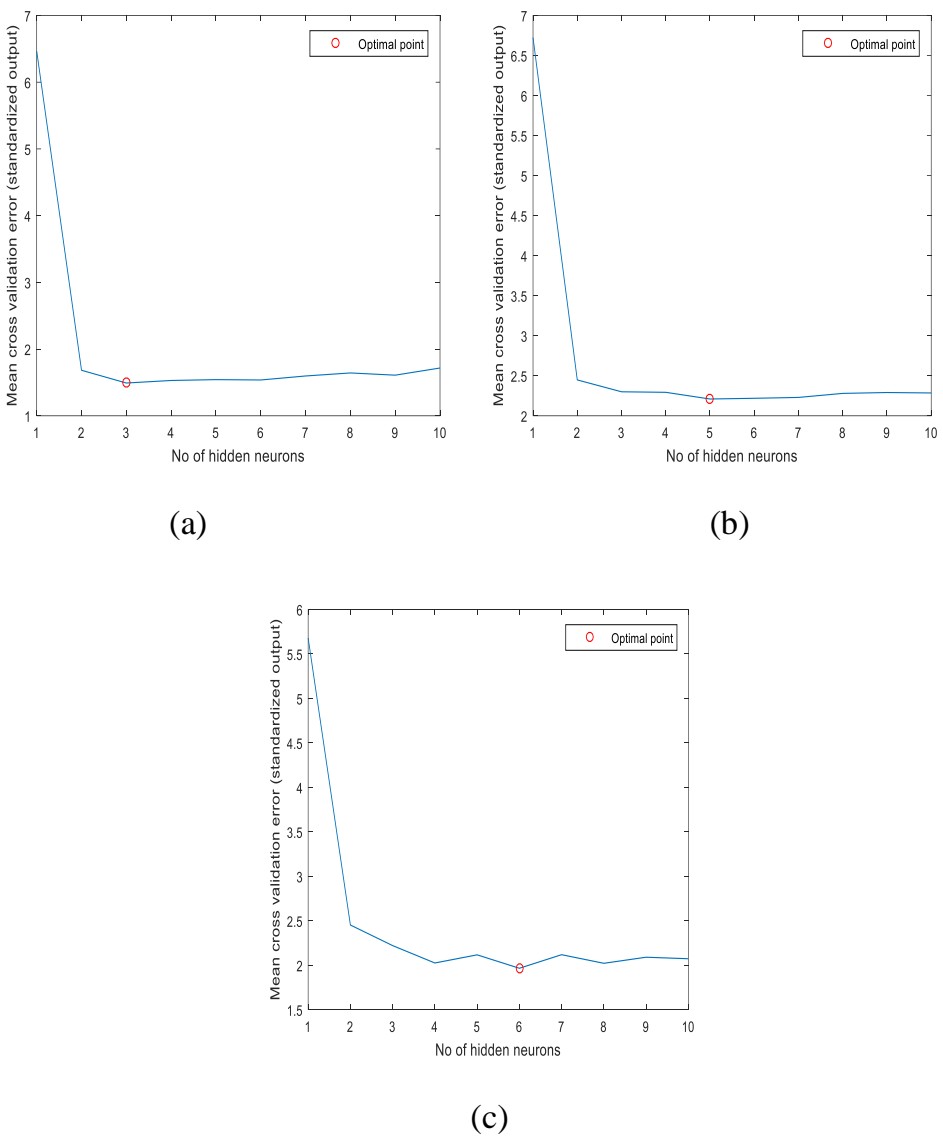

**Figure 3** **Optimal number of hidden neurons for: (A) station 1, (B) station 2, (C) station 3.**

where $\mathbf{I}$ represents the identity matrix, $\mathbf{k}_*$ represents a vector which can be defined by $[\mathbf{k}_*]_i = \mathrm{cov}(\boldsymbol{x}_i, \boldsymbol{x}_*)$, $k_* = \mathrm{cov}(\boldsymbol{x}_*, \boldsymbol{x}_*)$ and $\mathbf{K}$ represents a covariance matrix with elements $[\mathbf{K}]_{i,j} = \mathrm{cov}(\boldsymbol{x}_i, \boldsymbol{x}_j)$. Thus, it can be seen that compared to classical regression methods where only the parameters are used to determine the model prediction or output, and the model output depends on the training dataset $\boldsymbol{X}, \boldsymbol{y}$.

For precise predictions, the available data is used to determine the parameters of the mean and covariance function. The predictive probability distribution is completely defined using these parameters which are often referred to as hyperparameters. The values of the hyperparameters can be obtained by maximizing the log-likelihood function of the training

datasets (*Rasmussen & Williams, 2006*):

$$\log p(\boldsymbol{y}|\boldsymbol{X}) = -\frac{1}{2}\boldsymbol{y}^{\mathrm{T}}\left(\mathbf{K}+\sigma_n^2\mathbf{I}\right)^{-1}\boldsymbol{y} - \frac{1}{2}\log\left(\left|\mathbf{K}+\sigma_n^2\mathbf{I}\right|\right) - \frac{n}{2}\log(2\pi) \tag{6}$$

where $n$ is the number of training datasets.

### Bootstrap aggregated decision trees

In this paper, an ensemble of decision trees is also employed to model the daily river water temperature. A decision tree is a predictive modeling approach popular in machine learning. Specifically, for regression, it is referred to as regression tree. For regression trees, the tree structure is built by binary recursive portioning whereby the data is split into partitions or branches. All data in the training set are initially grouped into the same partition. Then the learning procedure assigns the datasets into the first two branches or partitions, by using every possible binary split of the inputs. The sum of the squared deviations from the mean in the two separate partitions is calculated and the split that minimizes this value is selected. This splitting procedure is then applied to each of the new partitions. The learning process continues until each node reaches a user-defined minimum node size and becomes a terminal node. An ensemble, which combines the predictions from multiple decision trees makes more accurate estimations than a single tree. Since decision trees are sensitive to the specific training data, their output has a high variance. In order to decrease the variance and increase accuracy, bootstrap aggregation or bagging is used (*Breiman, 1996*). Bagging creates several training datasets by using bootstrap sampling, i.e., by random sampling the original training set with replacement. The regression tree algorithm is applied to each dataset, and then the average amongst the models is used to calculate the estimations for the new data.

## Model evaluation

In this study, the following performance indicators were used to evaluate the model performance: the coefficient of correlation ($R$), and the root mean square errors ($RMSE$).

$$R = \frac{\sum_i(OV_i - OV_{\mathrm{mean}})(MV_i - MV_{\mathrm{mean}})}{\sqrt{\sum_i(OV_i - OV_{\mathrm{mean}})^2\sum_i(MV_i - MV_{\mathrm{mean}})^2}} \tag{7}$$

$$RMSE = \sqrt{\frac{\sum_{i=1}^n(MV_i - OV_i)^2}{n}} \tag{8}$$

where $OV_i$ is the observed water temperature, $OV_{\mathrm{mean}}$ is the averaged observed water temperature for the simulation time period, $MV_i$ is the simulated value, $MV_{\mathrm{mean}}$ is the averaged simulated value for the simulation time period, and $n$ is the number of daily estimated and corresponding observed water temperatures at a river monitoring station.

The Nash-Sutcliffe model efficiency coefficient ($NSC$), defined in the Eq. (9), was also used to evaluate the goodness of fit for each model. $NSC$ has a maximum value of 1.0 and has no minimum value. A value of 1.0 indicates a perfect model efficiency.

$$NSC = 1 - \frac{\sum_{i=1}^n(MV_i - OV_i)^2}{\sum_{i=1}^n(OV_{\mathrm{mean}} - OV_i)^2}. \tag{9}$$

**Table 2** Regression expressions of standard models for the three stations in this study.

| Station No. | Standard models | Expressions |
|---|---|---|
| 1 | Linear model | $T_w(t) = 6.088 + 0.288 \cdot T_a(t)$ |
| | Non-linear model | $T_w(t) = 1.606 + \frac{10.443}{1+e^{0.183(3.036 - T_a(t))}}$ |
| | Stochastic model | $T_w(t) = 7.149 + 6.661 \sin\left[\frac{2\pi}{365}(t + 578.734)\right] + 0.006 T_a(t) + 0.011 T_a(t-1) + 0.014 T_a(t-2)$ |
| 2 | Linear model | $T_w(t) = 4.971 + 0.666 \cdot T_a(t)$ |
| | Non-linear model | $T_w(t) = \frac{25.639}{1+e^{0.153(12.179 - T_a(t))}}$ |
| | Stochastic model | $T_w(t) = 8.998 + 8.402 \sin\left[\frac{2\pi}{365}(t + 516.282)\right] + 0.085 T_a(t) + 0.035 T_a(t-1) + 0.142 T_a(t-2)$ |
| 3 | Linear model | $T_w(t) = 3.311 + 0.839 \cdot T_a(t)$ |
| | Non-linear model | $T_w(t) = \frac{33.601}{1+e^{0.131(16.587 - T_a(t))}}$ |
| | Stochastic model | $T_w(t) = 10.105 + 8.748 \sin\left[\frac{2\pi}{365}(t + 376.967)\right] + 0.193 T_a(t) + 0.006 T_a(t-1) + 0.142 T_a(t-2)$ |

In addition to the above performance indicators, the normalized benchmark efficiency (BE) defined, in analogy to the NSC coefficient, in Eq. (10) was used to measure the performance improvement of a model over a given benchmark model (*Schaefli & Gupta, 2007*).

$$BE = 1 - \frac{\sum_{i=1}^{n}(MV_i - OV_i)^2}{\sum_{i=1}^{n}(OB_i - OV_i)^2} \qquad (10)$$

where $OB_i$ is the observed water temperature of the benchmark model.

# RESULTS AND DISCUSSION

The test dataset or validation dataset, in this paper was formed by selecting only the data from the year 2013. The remaining data from the original data set was used as the training or calibration dataset. The parameters of the developed models were determined using the training dataset. The test dataset, on the other hand, was used to determine the quality of each of the proposed models. The *RMSE* was used as the indicator to determine the optimal parameter values, while *R* and *NSC* were used to compare the model performances.

## Standard models

The parameters of the simple linear regression models, the nonlinear regression models and stochastic regression models were determined using the training dataset and the formulas are presented in Table 2.

The performance indicators of these models obtained for the testing datasets are shown in Table 3. The linear regression models, which describe the relationship between daily water and air temperature of the Missouri River have relatively strong correlation coefficients with $R = 0.91$ and $R = 0.92$ for station No. 2 and 3. However, for station No. 1, this relationship is relatively weak since $R = 0.62$. Generally, the linear regression model performed poorly for water temperature prediction, with *RMSE* values ranging from 3.41 to 3.94. The Nash-Sutcliffe coefficients ($NSC = 0.37, 0.82, 0.84$) indicate that the performance of the linear regression model might not be satisfactory. The results showed

**Table 3 Performance coefficients of the developed models by the testing datasets.**

| Station No. | Models | R | RMSE | NSC | BE |
|---|---|---|---|---|---|
| 1 | Linear model | 0.62 | 3.41 | 0.37 | – |
| | Non-linear model | 0.64 | 3.34 | 0.29 | – |
| | Stochastic model | 0.94 | 2.14 | 0.72 | – |
| | ANN | 0.9665 | 1.9471 | 0.7601 | 0.6432 |
| | GPR | 0.9664 | 1.9784 | 0.7523 | 0.6317 |
| | BA-DT | 0.9508 | 1.8777 | 0.7769 | 0.6682 |
| 2 | Linear model | 0.91 | 3.53 | 0.82 | – |
| | Non-linear model | 0.94 | 2.99 | 0.87 | – |
| | Stochastic model | 0.98 | 1.86 | 0.95 | – |
| | ANN | 0.9833 | 1.5561 | 0.9657 | 0.7852 |
| | GPR | 0.9840 | 1.5524 | 0.9658 | 0.7862 |
| | BA-DT | 0.9823 | 1.5826 | 0.9645 | 0.7778 |
| 3 | Linear model | 0.92 | 3.94 | 0.84 | – |
| | Non-linear model | 0.93 | 3.62 | 0.86 | – |
| | Stochastic model | 0.98 | 1.72 | 0.97 | – |
| | ANN | 0.9993 | 1.5649 | 0.9741 | 0.8594 |
| | GPR | 0.9897 | 1.4950 | 0.9764 | 0.8717 |
| | BA-DT | 0.9849 | 1.8072 | 0.9655 | 0.8125 |

that the daily water and air temperature may not be linearly correlated. The non-linear regression model performed a little bit better than linear regression model at station No. 2 and 3. However, for station No. 1, it performed poorly with *NSC* value being 0.29. A comparison of the performance coefficients of the stochastic model with those of the linear and non-linear models (Table 3) indicates that the stochastic model is a better one due to the higher values of *NSC* and *R* on one hand, and lower RMSE values on the other hand.

## Machine learning models

Models of the water-air temperature relationships were generated using ANN, GPR and BA-DT and compared to the standard models obtained in the previous subsection. The performance coefficients of all the proposed models are shown in Table 3.

Analyzing the data in Table 3 for the models obtained using machine learning methods, it can be noticed that these models outperform the standard models. They all have lower *RMSE* values and higher *R* and *NSC* values than the standard models. All three models obtained using machine learning methods have comparable results, with GPR having slightly better results for station No. 2 and 3, while BA-DT has slightly better results for station No. 1. All three models show high correlation coefficient values with the least value being $R = 0.9508$ for BA-DT. For station No. 1, the best model is BA-DT with RMSE $= 1.8777$, the correlation coefficient $R = 0.9508$ and NSC $= 0.7769$. For station No. 2, the best model is GPR with RMSE $= 1.5524$, and the correlation coefficient $R = 0.9840$ and NSC $= 0.9658$. For station No. 3, the best model is GPR with RMSE $= 1.4950$, and the correlation coefficient $R = 0.9897$ and NSC $= 0.9764$. These conclusions are further corroborated by the normalized benchmark efficiency coefficients determined only for the

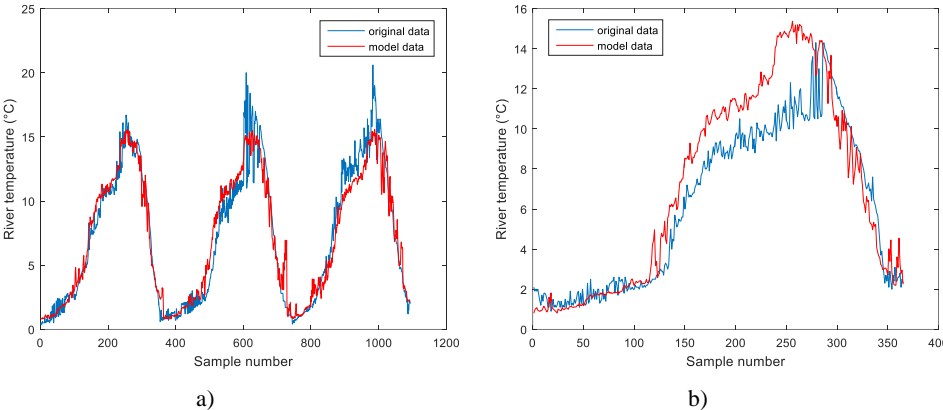

**Figure 4** Performance of BA-DT model for Station No. 1: (A) training dataset and (B) testing dataset.

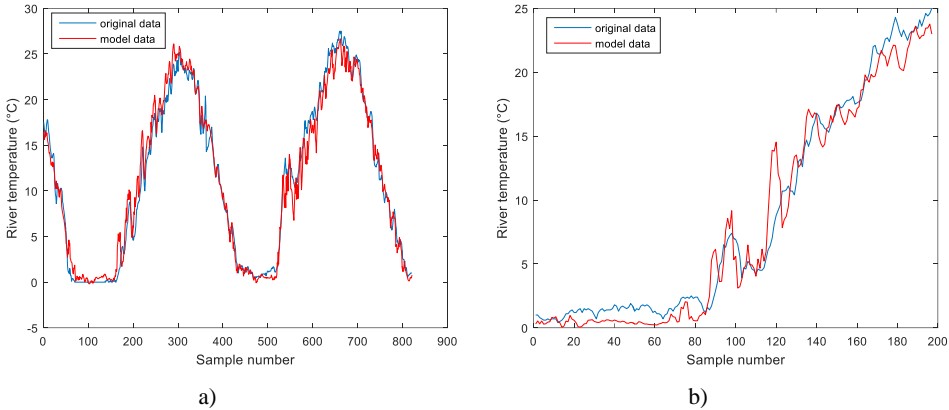

**Figure 5** Performance of GPR model for Station No. 2: (A) training dataset and (B) testing dataset.

machine learning models using the corresponding linear model as a benchmark model. A higher value indicates better performance improvement over the linear benchmark model compared to the other machine learning models.

The performance of the BA-DT model for station No. 1 is shown in Fig. 4, while the performance of GPR models for stations No. 2 and 3 are respectively presented in Fig. 5 and Fig. 6. It is shown that the BA-DT model performed poor in high water temperature period for station No.1 (Apple Creek). Apple Creek is a small tributary of the Missouri River, and it presents significantly different annual cycle of water temperature and flow discharge compared with stations No. 2 and 3 (Fig. 2). The poor model performance in station No. 1 is likely affected by flow discharge, which has not been included in the models used in this study. Flow discharge can significantly affect water temperature in many rivers, which has been reported by a lot of researchers (*Isaak et al., 2010*; *Van Vliet et al., 2013*; *Arismendi et al., 2014*; *Toffolon & Piccolroaz, 2015*; *Sohrabi et al., 2017*).

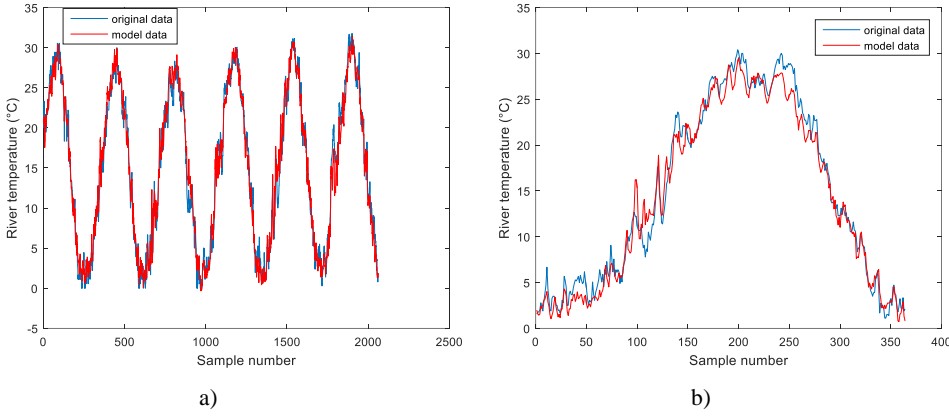

**Figure 6  Performance of GPR model for Station No. 3: (A) training dataset and (B) testing dataset.**

In general, standard regression models are easy to implement. However, their modelling performances are not good enough (Table 3). ANN models are easy to use, able to generalize better and can easily be adapted to include many inputs and outputs. However, it is a tough task to determine the optimal network structure. A decision tree is a predictive modeling approach popular in machine learning. Its implementation for water temperature modelling in this study reveals that this method can be applied for river water temperature modelling. GPR method performs well for water temperature modelling in the Missouri River. For the Missouri River Recovery Management Plan and Environmental Impact Statement Analysis, GPR approach and deterministic models can be integrated to better model water temperature in the river basin. Additionally, further study is needed to analyze the impact of flow discharge on water temperature prediction in the Missouri River.

## CONCLUSIONS

Although many factors influence the prediction of river water temperature, the objective of this study was to estimate the daily water temperature of the Missouri River with the aid of only the mean air temperature. In this paper, three standard models and three models obtained using machine learning procedures were used to predict the water temperature of the Missouri river at three locations. Analyzing the three standard models, the stochastic model clearly outperforms the standard linear model and a nonlinear model based on the logistic S-shaped function. On the other hand, all three machine learning models have comparable results and outperform the stochastic model. GPR performs slightly better at station No. 2 and 3, while BA-DT has slightly better results for station No. 1. All three models show high correlation coefficient values with the least value being $R = 0.9508$ for BA-DT. Machine learning models are powerful tools for the estimation of daily river temperature. The model results may be used to supplement missing water temperature data and support the deterministic water temperature models.

## ACKNOWLEDGEMENTS

We would like to thank all those who participated in this study. Many thanks to the Reviewers (Sebastiano Piccolroaz, Gnana Sheela and one anonymous reviewer) for taking the time to provide helpful and constructive feedback on how to improve the manuscript. We would also like to thank the Editorial Team for their time and comments, plus their guidance through this process.

### Funding

This work was jointly funded by the National Key R&D Program of China (2016YFC0401506), the Projects of National Natural Science Foundation of China (51679146, 51479120). The funders had no role in study design, data collection and analysis, decision to publish, or preparation of the manuscript.

### Grant Disclosures

The following grant information was disclosed by the authors:
National Key R&D Program of China: 2016YFC0401506.
The Projects of National Natural Science Foundation of China: 51679146, 51479120.

### Competing Interests

The authors declare there are no competing interests.

### Author Contributions

- Senlin Zhu and Marijana Hadzima-Nyarko conceived and designed the experiments, performed the experiments, analyzed the data, contributed reagents/materials/analysis tools, prepared figures and/or tables, authored or reviewed drafts of the paper, approved the final draft.
- Emmanuel Karlo Nyarko performed the experiments, analyzed the data, contributed reagents/materials/analysis tools, prepared figures and/or tables, authored or reviewed drafts of the paper, approved the final draft.

### Data Availability

   The raw data are provided in a Supplemental File.

### Supplemental Information

Supplemental information for this article can be found online at http://dx.doi.org/10.7717/peerj.4894#supplemental-information.

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
