# Peer review of "Modelling daily water temperature from air temperature for the Missouri River"

_PeerJ, doi:10.7717/peerj.4894_

## Round 0.1 · original submission · Major Revisions

Two reviewers recommend reconsideration of your paper following major revisions and the third one minor revision. I invite you to resubmit your paper after addressing all reviewer comments.

When resubmitting your manuscript, please carefully consider all issues mentioned in the reviewers' comments, outline every change made point by point, and provide suitable rebuttals for any comments not addressed.

·

Basic reporting

In this paper the authors compare the performance of three different machine learning models to predict river water temperature based on air temperature with that of three well-known and widely used statistical models. The case study is the Missouri River (USA), along which the authors selected three river and weather stations. The results show that the machine learning models outperform the statistical models, in terms of Root Mean Square Error (RMSE), Coefficient of Correlation (R), and Nash-Sutcliffe Efficiency Coefficient (NSC).

The manuscript is generally well organized and structured and it is well written. The authors should possibly improve the clarity of some paragraphs in section 2.3 where the machine learning procedures are described, in order to make them clearer also to readers that have no or little familiarity with this type of models.

While several fundamental works are acknowledged in the bibliography, I believe that others are missing and should be mentioned in the Introduction section. I provide a list of works that in my opinion are missing in the Comments for the author below. In general, I suggest reviewing the structure of the Introduction section as in its current form the logical flow could be improved in some parts (see Comments for the author below).

The authors provided the raw data used for the analysis, but they should also explicitly indicate the website where the data have been retrieved (see also my comments in Experimental design below). According to the Journal’s policy, I also believe that the scripts used for the modelling could be made available, possibly with a brief description of their usage.

Figures are esplicative, but I suggest modifying Fig 2 showing all stations. In fact, the air temperature series (i.e., the forcing of all models) is shown only in this figure, thus this information is not available for stations 1 and 2 (only station 3 is shown in Fig 2). In doing so, please improve the readability of the two series of data by reducing the dimension of the symbols or using a line. Possibly, showing also streamflow could be useful to better discuss the results, and in particular the low performance at station 1 (where I believe that streamflow plays a key role, see also my comments below). I suggest indicating the date instead of sample number in all figures with time series of water temperature. Probably Fig 1 could be simplified making it larger and indicating north, scale, and location of the stations directly in the left panel. Please, check the aspect ratio of the figure.

Experimental design

The research questions are well defined and are within the scope of the Journal.

The description of the methods could be improved in section 2.3, in order to make the paper clearer and accessible also to people which are not experts of machine learning techniques.

I ask the authors to revise the mode inter-comparison. The correct way to estimate the relative performance of the different models would be to use an information criterion, in order to properly account for the trade-off between the goodness of the results and the simplicity of the model (number of parameters). The authors should identify a fair criterion to be used when comparing machine learning and regression models, or otherwise comment on why this was not possible. If it is possible to univocally define the number of parameters in the proposed machine learning methods, an information criterion as e.g., BIC or AIC could be easily introduced in the analysis.

Please, consider also that given the strong seasonality river water temperature it would be better using a modified definition of the NSC, according to Schaefli and Gupta (2007). See also Piccolroaz et al., 2016, both for the use of this index and model inter-comparison based on an information criterion (AIC in that case).

Among the simple models to predict river water temperature based on air temperature, there is a third category of models characterized by a hybrid formulation that combines a physical derivation of the key equation with a stochastic calibration of parameters (Toffolon and Piccolroaz, 2015 air2stream model freely available at https://github.com/spiccolroaz/air2stream). I believe that the overall analysis would benefit if the authors could include also this type of models in their analysis as a model grounded on physical principles is missing. This is just a suggestion and not a request. I have no doubts that, if the authors decide to include it, they will be able to easily and competently add this model to the analysis (e.g., the 5-parameter version, which depends only on air temperature).

Validity of the findings

The time series of the three stations used by the authors have different duration. In particular, stations 1 and 2 have a much shorter time series for calibration and station 2 has an incomplete year for model validation. Model performance is certainly affected by the duration of the series used for calibration, making the comparison among the different stations unfair. I was wondering whether other data are available. Looking at the USGS website (https://waterdata.usgs.gov/nwis/), which is from where I believe the authors downloaded the data, I found that other stations are available along the Missouri River, some of those with much longer time series. I ask the authors to verify if other data are available for the stations that they used, and/or to add other stations with longer/comparable time series. At this link all stations in the Missouri region with available data are listed: https://waterdata.usgs.gov/nwis/dv?referred_module=sw&huc2_cd=10&site_tp_cd=ST&index_pmcode_00010=1&index_pmcode_00011=1&group_key=NONE&sitefile_output_format=html_table&column_name=agency_cd&column_name=site_no&column_name=station_nm&column_name=site_tp_cd&column_name=lat_va&range_selection=date_range&begin_date=2000-03-07&end_date=2018-03-06&format=gif&date_format=YYYY-MM-DD&rdb_compression=file&list_of_search_criteria=huc2_cd%2Csite_tp_cd%2Crealtime_parameter_selection (it takes some minutes to load the whole list).

I could not find water temperature for stations 1 and 3 in the USGS website, and station 2 covers a different period compared to that used by the authors. Please, make clear the source of data used in the analysis. In addition, station 1 is not along the main course of the Missouri River, but along a small tributary (Apple Creek). This explains the significantly different annual cycle of water temperature, and also the worse model performance. In fact, I believe that the shape of the temperature cycle of this river is likely affected by streamflow, a variable that is not included in the models used by the authors but that can significantly affect water temperature in many rivers (Isaak et al., 2010; van Vliet et al., 2011; Arismendi et al., 2014; Toffolon and Piccolroaz , 2015; Sohrabi et al., 2017). This should be explicitly commented in the discussion of the results. In this respect, adding also streamflow to Fig 2 could be useful.

The results part should be expanded commenting on why in some stations the models behave better than in others, why some models behaves better than others, what are the limitations of the models considered in the analysis (e.g., the effect of discharge). Notice also that the Missouri River is acknowledged for ice formation and floating, how this can affect the model predictions? It is also known that river water temperature is chiefly modulated by the hydrological regime of the river (Webb and Nobilis, 2007; Kelleher et al.2012; Lisi et al., 2015; Piccolroaz et al., 2016). What are the hydrological regimes of the Missouri River and of the Apple Creek? Can the authors prove or speculate the portability of the proposed models to rivers with other hydrological regimes? In this sense, considering the main course of the Missouri River and also some tributaries (e.g., the Apple Creek) could be of interest. Can the author comment of the possible extension of this type of model to regulated rivers (e.g., dammed rivers)?

In conclusion, I find the analysis proposed by the authors promising and potentially a valuable contribution in the field of water temperature modelling in rivers, but I ask to carefully consider the above comments before possible publication on this Journal.

Additional comments

Abstract: choose better wording than “creating” and “constructed”

Line 76: I would say that these are factor affecting river water temperature and not the “prediction” of river water temperature. Among these factors, acknowledge also anthropogenic impacts (Hester and Doyle, 2011) and the hydrological regime (Webb and Nobilis, 2007; Kelleher et al.2012; Lisi et al., 2015; Piccolroaz et al., 2016).

Line 81: remove “quite”

Line 84: add references, as e.g., Stefan and Preud'homme, 1993; Mohseni and Stefan, 1999; Caissie, 2006. Please, acknowledge also the limitations of using only air temperature, discharge may also play a crucial role (e.g., among others van Vliet et al., 2011; Arismendi et al., 2014; Toffolon and Piccolroaz , 2015).

Line 85: go to new line

Line 97-101: please, acknowledge also hybrid statistical-physical based models as air2water (Toffolon and Piccolroaz, 2015; Piccolroaz et al., 2016).

Line 116-119: it actually depends on the hydrological regime.

Line 119-122: see also (Stefan and Preud’homme (1993), Isaak et al. (2012), Toffolon and Piccolroaz (2015) for a discussion on the thermal inertia (depending on the hydrological regime of the river).

Line 124-125: please, reformulate.

Line 126-127: the annual component is the long-term component. What is missing here is the fluctuating short-term component. This is also discussed in Cassie, 2006.

Lines 131-132: please expand this sentence otherwise the 5th point is not very informative.

Line 133: what do the authors mean with “suitable lags”?

Line 138-140: people may do not know what are MLP, RBF, ANN, and GPR models. Probably this paragraph should start with an introductory sentence on machine learning models.

Line 149-151: the prediction of river flow is not the subject of the paper, so I would remove this sentence.

Line 133-161: this is a long paragraph that should be revised (see also the previous comments). The first sentence introduces the question of selecting appropriate model inputs (predictor variables) and lags (not sure what do the authors mean), but then this question is not answered in the coming sentences, where the focus shifts on model inter-comparison.

Section 2: please, provide information about the hydrological regime of the river, location of the stations (main course, tributaries). See also my previous comment about the use of additional stations. Add also all the necessary details about who provided the data and how they can be accessed.

Lines 196-197: … a non linear regression models and a stochastic regression model

Section 2.3: Do I correctly understand that the authors used t, Ta(t), Ta(t-1) and Ta(t-2) as predictors?
At line 230, Tw(t) should be water temperature not air temperature. I ask the authors to make the description of the machine learning models simpler to make it clearer for readers not familiar with this kind of models. A schematic of the three models could be helpful.

Equation 9: please, consider the use of the modified NSC, according to Schaefli and Gupta (2007)

Line 319-321: probably referring to calibration and validation periods would be better than “training dataset” and “test dataset”.

Table 3: do these values refer to the training (calibration) or test (validation) period? From the caption of the table I understand that they refer to the testing (validation) period. Is this correct? I would probably show all performance metrics for both periods.

Line 375: the possibility to easily include additional input variables is an important statement. Probably this goes beyond the scope of the current analysis, but an interesting extension of the study could be adding also streamflow in input and compare the three machine learning models with other models that use both air temperature and streamflow (Toffolon and Piccolroaz , 2015; Sohrabi et al., 2017). This could be commented in the conclusions or at the end of the results section.

Lines 378-380 are a repetition of lines 263-265

Section 3: I feel like in this section a detailed discussion of results is missing. I would rename the section as Results and Discussion and add some detailed comments on the results. See also my previous comments in “Validity of the findings”

Bibliography

Arismendi I, Safeeq M, Dunham JB, Johnson SL. 2014. Can air temperature be used to project influences of climate change on stream temperature? Environmental Research Letters 9: 084015. DOI:10.1088/1748-9326/9/8/084015

Caissie D. 2006. The thermal regime of rivers: a review. Freshwater Biology 51: 1389–1406. DOI:10.1111/j.1365-2427.2006.01597.x

Hester, E. T., and M. W. Doyle (2011), Human impacts to river temperature and their effects on biological processes: A quantitative synthesis, JAWRA Journal of the American Water Resources Association, 47 (3), 571{587, doi:10.1111/j.1752-1688.2011.00525.x.

Isaak, D. J., C. H. Luce, B. E. Rieman, D. E. Nagel, E. E. Peterson, D. L. Horan, S. Parkes, and G. L. Chandler (2010), Effects of climate change and wildfire on stream temperatures and salmonid thermal habitat in a mountain river network, Ecological Applications, 20 (5), 1350{1371, doi:10.1890/09-0822.1.

Isaak, D. J., S. Wollrab, D. Horan, and G. Chandler (2012), Climate change effects on stream and river temperatures across the northwest u.s. from 1980{2009 and implications for salmonid fishes, Climatic Change, 113 (2), 499{524, doi:10.1007/s10584-011- 0326-z.

Kelleher C, Wagener T, Gooseff M, McGlynn B, McGuire K, Marshall L. 2012. Investigating controls on the thermal sensitivity of Pennsylvania streams. Hydrological Processes 26: 771–785. DOI:10.1002/hyp.8186

Lisi, P. J., D. E. Schindler, T. J. Cline, M. D. Scheuerell, and P. B. Walsh (2015), Watershed geomorphology and snowmelt control stream thermal sensitivity to air temperature Geophysical Research Letters, 42 (9), 3380{3388, doi:10.1002/2015GL064083.

Mohseni O, Stefan HG. 1999. Stream temperature/air temperature relationship: a physical interpretation. Journal of Hydrology 218:128–141. DOI:10.1016/S0022-1694(99)00034-7

Piccolroaz, S., E. Calamita, B. Majone, A. Gallice, A. Siviglia, and M. Toffolon (2016), Prediction of river water temperature: a comparison between a new family of hybrid models and statistical approaches, Hydrological Processes, 30 (21), 3901{3917, doi:10.1002/hyp.10913, hYP-16-0208.R1.

Schaefli B, Gupta HV. 2007. Do Nash values have value? Hydrological Processes 21: 2075–2080. DOI:10.1002/hyp.6825

Sohrabi, M. M., R. Benjankar, D. Tonina, S. J.Wenger, and D. J. Isaak (2017), Estimation of daily stream water temperatures with a bayesian regression approach, Hydrological Processes, 31 (9), 1719{1733, doi:10.1002/hyp.11139, hYP-15-0782.R2.

Stefan HG, Preud’homme EB. 1993. Stream temperature estimation from air temperature. Journal of the American Water Resources Association 29: 27–45. DOI:10.1111/j.1752-1688.1993.tb01502.x

Toffolon M, Piccolroaz S. 2015. A hybrid model for river water temperature as a function of air temperature and discharge. Environmental Research Letters 10: 114011. DOI:10.1088/1748-9326/10/11/114011

van Vliet, M. T., W. H. Franssen, J. R. Yearsley, F. Ludwig, I. Haddeland, D. P. Lettenmaier, and P. Kabat (2013), Global river discharge and water temperature under climate change, Global Environmental Change, 23 (2), 450 { 464, doi:https://doi.org/10.1016/j.gloenvcha.2012.11.002.

Webb, B. W., and F. Nobilis (2007), Long-term changes in river temperature and the influence of climatic and hydrological fact

Reviewer 2 ·

Basic reporting

Language and text was in good shape, and I thought the background information was adequate. I think the figures are ok, but styles differ throughout (figure 2 looks like a excel screenshot) and it isn't clear why only certain models are included in the evaluation figures 4-6. We only see timeseries for two of the three ML models and none of the traditional models. A number of citations were dated but referred to as "recent work" (see more below). The examples for ecosystem value of river temperature data was limited and I think the authors shouldn't have their first example be a coral reef since this paper is about river temperatures. I don't see the source of the data cited. USGS data should be cited as "U.S. Geological Survey, 2018, National Water Information System—Web interface, accessed {date}, at http://dx.doi.org/10.5066/F7P55KJN.";

Experimental design

The research design is adequately described and we have enough details to reproduce it. I do not think the authors went far enough to make this compelling. Their study area was a rare situation were they have good historical USGS temperature data and did not evaluate two things that would make these results more useful to the discipline: 1) impact of missing/sparce training data (the authors have a lot of training data, as opposed to evaluating the various models when 50% are removed, or more). 2) the ability to use one station's temperature data as input into another station model. Temperature stations could be leveraged in the modeling approach since temperature is the most widely measured aquatic measurement. What is the impact of including up/downstream measurements in various model formulations. I think the authors either need to show their results (the ranking of models) are appropriate elsewhere in the very different climate/flow regime, or add more depth to their study (such as the two suggestions above).

Validity of the findings

The findings seem valid but it is challenging to understand how transferable the results are. It isn't clear why the authors only use three sites and if the results could be applied elsewhere. We also could benefit from seeing/discussing limitations of the various approaches, such as seasonal biases or prediction failures. It is also odd to have the test dataset for station 2 to use an incomplete year. This makes it so we can't evaluate later season predictions. Please change than and use a full season for test. It isn't clear why figures 4-6 only represent two ML models, and do not (in one case, figure 5) show the model w/ the best performance (ANN in the case of station 2).

Additional comments

line 67: don't use a coral reef fish temperature example in a paragraph about river temperature. Use a river temperature example (there are many)
lines 62-72: also should mention the relationship between gasses and temperature solubility, which impacts ecosystem health. Also, temperature-dependent metabolism.
line 73: why is it a complex issue "to accurately predict water temperature"? tell us why or provide a citation. These things are all measureable, doesn't that make it easier? Isn't the conclusion of the paper that this is a relatively simple issue?
line 84: also air temperature is widely measured and more available than full energy budget components.
line 87: the Caissie and Benyahya papers are over ten years old.
line 89: "mass balances of flow fluxes in a water body" note these water fluxes also contribute to the energy/heat budget
line 92: "tremendously employed" what does that mean?
line 157: revise: "In some cases, ANN models provided better results, while in other some other regression models."
line 319: more info needed for the dataset. What was the remaining data? What time periods are available? Why were only three sites used?
line 337: is RMSE "daily" or instantaneous?
figure 5: test data don't include the whole season for station 2
table 1: should include date ranges

·

Basic reporting

NO COMMENTS

Experimental design

NO COMMENTS

Validity of the findings

NO COMMENTS

Additional comments

The paper is presented well, and also acceptable for publication.The following suggestion is as follows,
1.Include more recent papers inliterature review
2.how to find optimal no. off hiddden neuron in the proopsed model
33. More explqnation is required in the comparative analysis of ANN. GPR .BA DT models

---

## Round 0.2 · accepted · Accept

After careful analysis of the revision of the article, I consider that it has improved greatly and now can be published in the journal. Congratulations!